# Size Effects in Optical and Magneto-Optical Response of Opal-Cobalt Heterostructures

**DOI:** 10.3390/ma14133481

**Published:** 2021-06-22

**Authors:** Irina A. Kolmychek, Ksenia A. Lazareva, Evgeniy A. Mamonov, Evgenii V. Skorokhodov, Maksim V. Sapozhnikov, Valery G. Golubev, Tatiana V. Murzina

**Affiliations:** 1Physics Department, M. V. Lomonosov Moscow State University, Leninskie Gory, 119991 Moscow, Russia; ya.ksenia-lazareva@yandex.ru (K.A.L.); mamonov@shg.ru (E.A.M.); murzina@mail.ru (T.V.M.); 2Institute for Physics of Microstructures, RAS, 603950 Nizhny Novgorod, Russia; evgeny@ipmras.ru (E.V.S.); msap@ipmras.ru (M.V.S.); 3Radio-Physical Department, Lobachevsky State University of Nizhny Novgorod, 603950 Nizhny Novgorod, Russia; 4Ioffe Institute, Politekhnicheskaya ul. 26, 194021 St. Petersburg, Russia; golubev.gvg@mail.ioffe.ru

**Keywords:** synthetic opal, photonic bandgap, surface plasmon-polariton, magneto-optics

## Abstract

Search for new types of efficient magnetoplasmonic structures that combine high transparency with strong magneto-optical (MO) activity is an actual problem. Here, we demonstrate that composite heterostructures based on thin perfectly-arranged opal films and a perforated cobalt nanolayer meet these requirements. Anomalous transmission appears due to periodic perforation of Co consistent with the regular set of voids between opal spheres, while resonantly enhanced MO response involves the effects of surface plasmon-polariton (SPP) excitation at opal/Co interface or those associated with photonic band gap (PBG) in opal photonic crrystals. We observed the enhancement of the MO effect of up to 0.6% in the spectral vicinity of the SPP excitation, and several times less strong effect close to the PBG, while the combined appearance of PBG and SPP decreases the resultant MO response. Observed resonant magneto-optical properties of opal/Co heterostructures show that they can be treated as functional self-assembled magnetoplasmonic crystals with resonantly enhanced and controllable MO effect.

## 1. Introduction

Functional metamaterials designed for the controllable manipulation over the parameters of light have attracted a lot of interest during the last few decades. Modern achievements in the manufacturing technologies allow for the precise production of nanostructures made of different materials and with required parameters that can find a plenty of applications in nanophotonics, sensorics and optical switching. Plasmonics traditionally plays an important role in the field of light-matter interaction due to special plasmon-resonance-assisted amplitude and phase relations between the incident and outgoing electromagnetic field [1]. Surface plasmon polaritons (SPP) are the propagating electromagnetic excitations localized at a planar metal/dielectric interface, with the amplitudes of the electric field decaying exponentially inside both adjacent media [2,3]. SPPs dispersion dictates larger value of its wavevector as compared with that of a photon in the free space; thus, special efforts are required for the fulfillment of the momentum conservation for the SPP excitation. Phase mismatch can vanish in the case of prizm schemes or can be compensated by means of the reciprocal lattice vector in periodic structures; thus, metal/dielectric gratings are traditionally used [3]. SPP excitation in a 1D or 2D periodic array of metal nanostructures can be detected as a transmission minimum with the spectral position depending on the angle of incidence, θ [4,5].

Plasmonic metasurfaces are commonly made by different lithographic techniques that are relatively expensive and complicated in operation. Thus, the idea to form resonant SPP structures using self-assembled periodic dielectric structures supplemented by thin metal layer is rather attractive. Such an approach was realized for self-assembled colloidal monolayers covered by a metal nanofilm, so that periodic hexagonal array of nanocaps and nanoholes is formed [6,7,8,9]. Moreover, in this case, nanoperforated corrugated metal film supports the excitation of SPP, while the existence of nanoholes can lead to anomalous transmission instead of resonant absorption intrinsic to continuous plasmonic structures. The excited plasmons lead to an increase in the evanescent field near the film surface, and, after this enhanced field tunnels through the holes, the reverse transformation of the surface plasmons into the radiation field occurs [10,11]. This opens a way for efficient applications of such structures.

Photonic crystals (PhC) are another type of nanostructured photonic dielectric material; its permittivity is periodically modulated on the length scale comparable with the light wavelength. Multiple interference of the electromagnetic waves scattered from each unit cell of a PhC results in the formation of the so-called photonic bandgap (PBG), which is the spectral range of electromagnetic waves that cannot propagate through the PhC structure [12]. This phenomenon is of a great fundamental and practical significance which allows to use PhCs in lasers, light-emitting diodes, highly efficient waveguides, high-speed optical switches, and sensors [12], as well as for the enhancement of nonlinear-optical effects [13,14]. The PBG appears as a maximum in the reflection spectrum or as a minimum in the transmission one, its spectral position being a function of the angle of incidence. A well known type of 3D PhC is artificial opal consisting of closely packed SiO${}_{2}$ colloidal spheres; for the spheres’ diameter of several hundreds of nanometers the PBG corresponds to the visible range, with the spectral position and width being determined by the diameter of the particles and the interparticle filling [15].

Silver-opal heterostructure providing both SPPs and PBG was realized in References [16,17]. Despite the fact that the lattice periods of the PC and plasmonic crystals are the same, the spectral features of these crystals differ due to different dispersion laws of photons and SPPs. It was found that the spectrum of silver-opal heterostructure demonstrates both anomalously high SPP-assisted transmission and its attenuation due to the PBG. At the same time, detailed studies of the interplay of these features is still missing.

Additional control over the parameters of light can be provided by static magnetic field if a ferromagnetic material is installed in a nanostructure. Periodic magneto-optical (MO) nanostructures that support the SPP excitation are recalled magnetoplasmonic crystals (MPCs) and were studied in a number of papers. The investigated structures were obtained both by different top-down lithographic techniques [4,5,18,19,20,21,22] and by the bottom-up self-assembly method, such as metal deposition on the surface of colloidal crystal [23].

Magneto-optical (MO) effects appearing as polarization plane rotation usually possess significant enhancement in the spectral vicinity of the SPPs excitation due to the selective suppression of one of the components of the electric field of light. The enhancement of intensity MO effects (transversal Kerr and Voigt) are caused by the magnetic-field-induced spectral shift of the SPP dispersion curve [4]. Anyway, the MO effects’ amplification is usually observed in the spectral vicinity of the transmission damping.

As for the PhCs made in part of MO materials and known as magnetophotonic crystals, the combination of gyrothropy and PBG was firstly realized in 1D PhC based on Bi:YIG and nonmagnetic dielectrics [24]; such PhC revealed an enhancement of the Faraday effect accompanied by the decrease of transmittivity, which restricts its application in transmission geometry. This problem can be overcome using different approaches, like special order of alternating magnetic and nonmagnetic layers of the 1D PC, realized in References [12,25]; 2D magneto-photonic crystals are usually constructed as arrays of nanorods or nanopoles of Bi:YIG or a ferromagnetic metal [26,27,28], while 3D magneto-photonic crystals often represent either opal-like structures, where the interparticle space is filled with magneto-active medium (i.e., magnetite Fe${}_{3}$O${}_{4}$ [29], Faraday-active liquid glycerol solution of dysprosium nitrate [30]), or magnetic inverse-opal slabs [31,32]. Magnetophotonic crystals based on opals infiltrated by a ferromagnetic specimen [29] and opal/Bi:YIG/opal heterostructures [33] are more perspective for the application in the reflection geometry, as, for a narrow spectral range near the PBG, large reflectivity is accompanied by the enhancement of the Kerr rotation, along with the change of its sign and of magnetization-induced circular dichroism.

In this paper, we study optical and magneto-optical (MO) response of composite magnetophotonic structures based on opal films supplemented by a perforated cobalt nanolayer. We show that, depending on the silica spheres’ diameter, such composites demonstrate enhanced MO effect induced by the surface plasmon polaritons’ excitation or by photonic bandgap effect, along with anomalous transmission.

## 2. Materials and Methods

In this study, we used specially synthesized opal films composed of monodisperse amorphous silica (a-SiO${}_{2}$) colloidal spheres forming the hexagonal lattice with the (111) plane parallel to the substrate surface. The fabrication of the films consisted of two stages. First, monodisperse spherical a-SiO${}_{2}$ particles were produced by slow alkaline hydrolysis of tetraethoxysilane in an aqueous-alcoholic medium. To improve the thermal and chemical stability, a-SiO${}_{2}$ spheres were subjected to supplementary thermal annealing at T = 900${}^{\circ }$. At the second stage, 3D-ordered opal films containing 10–15 monolayers were grown from water suspension of a-SiO${}_{2}$ spheres by liquid phase colloid epitaxy on 0.5-mm-thick fused silica wafers [34].

Opal films with the spheres’ diameter *d* = 250, 310, 370, 520, and 640 nm were made, denoted below by the numbers “1”, “2”, “3”, “4”, and “5”, respectively. Then, a thin cobalt layer was deposited on the (111) opal surface by using the magnetron sputtering, the effective thicknesses of Co being $h_{eff}\approx 0.1d$. Thus, the SiO${}_{2}$ particles have been hemispherically covered and the samples can be treated as containing 2D periodic hexagonal arrays of nanoholes between the hemispheres, while the individual caps are connected. Reference flat cobalt films with the same effective thicknesses were deposited on a glass substrate in the same process. The real average thickness of the cobalt layer covering the opal spheres’s surface <h> is approximately twice less than that of the same mass thickness deposited on the flat surface, as the surface area of a hemisphere is twice the area of the circle on which it rests. The morphology of the samples has been studied by scanning electron microscopy (Figure 1a, upper inset); one can see a good periodicity and dense package of the structure.

Magnetic characterization of the samples was carried out by means of the longitudinal magneto-optical Kerr effect (MOKE) sketched in Figure 1b. It should be noted that the surface of the opal is formed by crystallites with different in-plane orientations of the crystal axes. The typical crystallite size is 10–40 μm. The diameter of the light beam is about 150 μm; thus, the MO response is averaged over several crystallites with different orientations. Figure 1c–g shows the MOKE hysteresis loops obtained for the studied structures demonstrating a waist near H=0, which is consistent with the formation of vortex magnetization observed previously for similar composites based on colloid PMMA crystals [35,36]. For the heterostructures “4” and “5”, zero width of the loop near H=0 indicates that all the magnetic hemispheres of the sample exhibit vortex magnetization state, while, for other samples, a constriction of the magnetic hysteresis loops at H=0 is the evidence of a mixed magnetization distribution, i.e., both Co hemispheres in the vortex state and in the single-domain state exist in the system. According to the expression $\frac{M_{sat}-M_{res}}{M_{res}}$, where $M_{sat}$ and $M_{res}$ are saturated and residual magnetizations, the relative amount of particles in vortex magnetization state are about 45, 70, and 80% for the samples “1”, “2”, and “3”, respectively. At the same time, reference cobalt film on flat glass substrate revealed a typical rectangular-like magnetic hysteresis loop similar to that shown, e.g., in Reference [36]. The saturated magnetic field for all the structures in the in-plane direction does not exceed 1 kOe. The detailed influence of the morphology of the nanocaps covering the colloidal crystal on the magnetization loops was presented in Reference [37].

Wavelength-angular transmission optical spectra T(θ,λ) were studied using a setup based on a halogen lamp as a broadband light source; the spectra were normalized to that for the corresponding reference flat cobalt film, in order to reveal the possible anomalous transmission. For the MO experiments, the static magnetic field of *H* = 2 kOe oriented perpendicular to the plane of incidence was applied to the heterostructures (Voigt geometry), as shown in the inset in Figure 2l. Accordingly, to the MOKE results, this field is enough for the in-plane saturation of magnetization. To evaluate the MO response of the sample, we measured the magnetic contrast in a wide range of the pump wavelengths and incident angles, defined as $\rho \left(\theta ,\lambda \right)=\frac{T_{+}\left(\theta ,\lambda \right)-T_{-}\left(\theta ,\lambda \right)}{T_{+}\left(\theta ,\lambda \right)+T_{-}\left(\theta ,\lambda \right)}$, where $T_{+}\left(\theta ,\lambda \right)$ and $T_{-}\left(\theta ,\lambda \right)$ are the transmission coefficients obtained for the opposite directions of the applied magnetic field.

## 3. Experimental Results

The experimental results are shown in Figure 2. For the samples “1” and “2”, there is only one clear spectral feature, which appears as an arc-like minimum at λ ∼ 500–550 nm (Figure 2a) and λ ∼ 550–600 nm (Figure 2c), respectively. In the case of the sample “3”, there is a V-shaped region with an anomalous transmission (red-yellow-colored area) centered at 460 nm at normal incidence, where the normalized transmission reaches the value of 3 (Figure 2e). There is also a less pronounced arc-like gap located near the wavelength of 700 nm. For the samples “4” and “5”, the T(θ,λ) spectra demonstrate V-like regions of anomalous transmission, where the value of T achieves 4 and 12, respectively (Figure 2g,i). It should be noted that the relative surface area of nanoholes is the same for all the samples and is less than 9%.

The ρ(θ,λ) dependencies are odd with respect to the angle of incidence that is typical for the Voigt configuration (Figure 2b,d,f,h,j). The ρ orders of magnitude are expectable for magneto-plasmonic structures [4,38]. In all the ranges, the data demonstrate a complicated sign-alternating behavior, the physical mechanism of which will be discussed below. The wavelength-angular spectrum of a flat cobalt film is monotonous and of permanent sign for θ>0, and the values of the magnetic contrast are about 5$\cdot 10^{-4}$.

## 4. Calculations

For the calculations of the SPP dispersion curves, we used a model of planar 2D grating bounded by opal and air semi-infinite regions [16]. In our samples, the cobalt layer thickness is comparable with the skin depth; thus, Co/opal and Co/air are not independent interfaces [5,39]. Based on the Maxwell equations and boundary conditions, the following equation for the dispersion of the bound SPP modes excited in a thin metal film squeezed between two dielectrics was obtained [3]:(1)$$\begin{array}{c} e^{-2k_{1}<h>}=\frac{k_{1}/\epsilon_{1}+k_{2}/\epsilon_{2}}{k_{1}/\epsilon_{1}-k_{2}/\epsilon_{2}}\frac{k_{1}/\epsilon_{1}+k_{3}/\epsilon_{3}}{k_{1}/\epsilon_{1}-k_{3}/\epsilon_{3}}\\ k_{i}^{2}=\beta^{2}-k_{0}^{2}\epsilon_{i},i=1,2,3, \end{array}$$
where <h> is the cobalt thickness, $\vec{\beta }$ is the SPP wave vector, ${\vec{k}}_{0}$ is the wave vector of light in vacuum, and $\epsilon_{1}$, $\epsilon_{2}$, and $\epsilon_{3}$ are the dielectric functions of cobalt, opal, and air, respectively. The effective epsilon of the opal was calculated as $\epsilon_{2}=\epsilon_{SiO_{2}}*f+\left(1-f\right)$, where *f* is the volume fraction of the spheres [40]. The data of the optical properties of bulk cobalt and silicon dioxide were taken from Reference [41].

Momentum conservation law for a 2D plasmonic crystal takes the following form [10]:(2)$$\begin{array}{c} \vec{\beta }={\vec{k}}_{x}+m{\vec{G}}_{1}+n{\vec{G}}_{2}, \end{array}$$
where $k_{x}=k_{0}sin\theta$ is the in-plane component of the wave vector of the incident light, (m,n) is a pair of integers and $|{\vec{G}}_{1}\left|=\right|{\vec{G}}_{2}|=\frac{4\pi }{\sqrt{3}d}$ are the reciprocal hexagonal lattice vectors for the opal consisting of spheres with the diameter *d*. By squaring the Equation (Equation 2), substituting the expression for $\beta^{2}$ to (Equation 1) and numerically solving the transcendental equation in the Wolfram Mathematica package, the SPP dispersion curves were obtained, shown in Figure 2a–j by solid and dashed curves. Corresponding pairs of integers (m,n) are indicated near the curves. As expected, for every pair (m,n), there are two modes of the SPP: upper and lower branch, corresponding to the short-range and long-range excitations, respectively [3] (indicated as “s” and “l” in Figure 2a–j).

The penetration depth into the i-th medium of the SPPs propagating at the interface between i-th and j-th media can be estimated as $l_{i}=\frac{1}{k_{0}}\left|\frac{\sqrt{-(\epsilon_{i}+\epsilon_{j})}}{\epsilon_{i}}\right|$ [3], the obtained value for cobalt $l_{1}$ is about 20–25 nm, i.e., it is indeed comparable with the layer thickness. For the opal and air, the corresponding values are $l_{2}$ ∼ 350–400 nm and $l_{3}$ ∼ 600–650 nm, respectively.

The spectral position of the PBG central wavelength was calculated as [40]:(3)$$\begin{array}{c} \lambda_{PBG}\left(\theta \right)=\sqrt{\frac{8}{3}}d\sqrt{\epsilon_{2}-sin^{2}\theta }, \end{array}$$
and the result is shown in Figure 2a–f by thick black line. For the samples “4” and “5” with *d* = 500 and 640 nm, the PBG is outside the spectral range under study.

It should be noted that different dependencies of the spectral positions of SPPs’ modes and $\lambda_{PBG}$ on the spheres’ diameter *d* allows to analyze optical properties of the heterostructures associated with mutual influence of these two features.

## 5. Discussion

Comparison between the experimental and theoretical data allows to conclude that arc-like transmission minima corresponds to the PBG in opal films. It is the most pronounced for the sample “1”, with the smallest diameter of spheres.

V-like transmission maxima in the spectra of the heterostructures “3”, “4”, and “5” apparently correspond to the (1,0) and (−1,0) SPPs excitations as they possess the same slope. The effect of extraordinary transmission in opal-cobalt heterostructure is associated with the tunnelling of the electromagnetic field through the holes via the SPPs excitation, as it was observed in different planar subwavelength holes’ arrays [10] and metal-coated colloidal monolayers [6,7]. The spectral shift of anomalous transmission with respect to the calculated SPP dispersion curves can be associated with the scattering losses in subwavelength nanoholes [42,43], non-planar shape of the perforated film, complicated shape of the holes, redistribution of intensity between different diffraction orders [10], etc. It should be noted that the long-range (1,0) and (−1,0) modes match well with the local transmission minimum, analogously to the spectra of cylindrical nanoholes’ arrays in Ag/Py film [21] and analogous apertures’ array in Au film [43]. Spectral positions of V-like transmission maxima are closer to the short-range branches than to long-range ones. The SPP excitations are readily scattering back to electromagnetic waves. The latter emerge from both sides of the sample and, thus, contributes to the enhanced transmission [17]. At the same time, optical spectra of the samples “1” and “2” do not reveal the regions of anomalous transmission in the range under study, which should include the SPPs resonances according to calculations. The enhanced transmission in this case is probably compensated by the PBG-assisted transmission gap observed close to the SPPs in these samples. The long-wavelength regions with high transmission are associated with low scattering of light in this spectral range.

Nonzero magnetic contrast for a planar ferromagnetic film, which corresponds to the existence of linear in magnetization intensity effect in the Voigt geometry, can appear due to the asymmetry of its boundaries [44]. In opal-cobalt heterostructures in the (θ,λ) regions far from PBG and SPPs, there is also an asymmetry of opal versus air. Near the SPPs excitations the enhancement of magnetic contrast is associated with the magnetization-induced spectral shift of the SPP dispersion curve towards the opposite directions when applying the magnetic fields +H and −H. It provides the zero magnetic contrast and the sign change of ρ in the spectral vicinity of the SPP excitation and increase in |ρ| on the sides of the SPP dispersion curve. The magnetic contrast growth is more pronounced near the long-range branches of the SPP curves as compared to those near the short-range ones. A clear sign modulation of ρ at the (1,0) and (−1,0) SPP lines is observed in all the structures under study except the sample “1”. The resonant maximum values of the magnetic contrast magnitude increase with increasing cobalt thickness that is typical for the linear MO effects.

It is of crucial importance that the angular-wavelength spectra of the samples “3”, “4”, and “5” contain the regions, where both transmittance and magnetic contrast are enhanced. In order to illustrate this fact, the cross-sections of T(θ,λ) and ρ(θ,λ) for $\theta =-10^{\circ }$ for the sample “4” are shown together (Figure 2k). The wavelengths corresponding to the SPPs excitation for this angle of incidence are indicated by vertical lines. It can be seen that the anomalous normalized transmission of up to 3.9, along with the enhancement of the magnetic contrast up to $10^{-3}$, are observed in the spectral range λ∈(680;760) nm. This combination of properties followed by a relatively sharp fronts of the ρ spectrum is perspective for nanophotonics and light manipulation devices.

Unfortunately, due to restricted spectral range available in our experimental setup, it is impossible to study the spectral features of ρ in the spectral vicinity of PBG for the samples “4” and “5”. For the sample “3”, a slight enhancement of |ρ| on the long-wavelength side of the PBG is observed that is probably due to the transmission gap, i.e., to the denominator minimum in the expression of the magnetic contrast. Moreover, there is no sign change associated with the PBG for these composite structures.

For the heterostructures “1” and “2”, the sign change of magnetic contrast, along with its enhancement on only one side, occurs near the PBG. It can be clearly seen at λ∼600 nm in the cross-sections of T(θ,λ) and ρ(θ,λ) for $\theta =15^{\circ }$ in the sample “2” (Figure 2l). The position of (−1,0)s SPP is at about 540 nm for this θ (grey dashed curve). It should be reminded here that: (i) there is no pure-PBG-assisted sign change of ρ in the sample “3”; and (ii) pure SPP-assisted behavior of magnetic contrast implies the sign change, along with the local amplification of |ρ|, on *both* long- and short-wavelength regions adjacent to zero ρ value. Based on these considerations, we can conclude that, in the samples “1” and “2”, both features, i.e., PBG and SPP, play a role in the spectral behavior of the magnetic contrast, despite the fact that spectral position of SPPs and PBG are different. The exact mechanism underlying the described effects in the considered heterostructures is a subject for future studies.

## 6. Conclusions

In conclusion, we have experimentally studied optical and magneto-optical properties of a set of magneto-optical opal/Co-based composites that support the SPPs’ excitations and the appearance of the PBG. SPP-assisted extraordinary transmission and a significant enhancement of the magneto-optical response in the Voigt geometry for the heterostructures formed by the SiO${}_{2}$ spheres’ with the diameters larger than 400 nm are observed simultaneously in the same spectral range, which is promising for nanophotonics applications. It is demonstrated that, for the heterostructures with close spectral positions of SPPs and PBG, their combined effect governs the magneto-optical behavior. Unique optical properties demonstrated by opal/Co heterostructures allow to suggest them as high-transparent self-assembled composites with a variety of resonant MO features competitive with magnetoplasmonic crystals made by lithography techniques.

## Figures and Tables

**Figure 1 materials-14-03481-f001:**
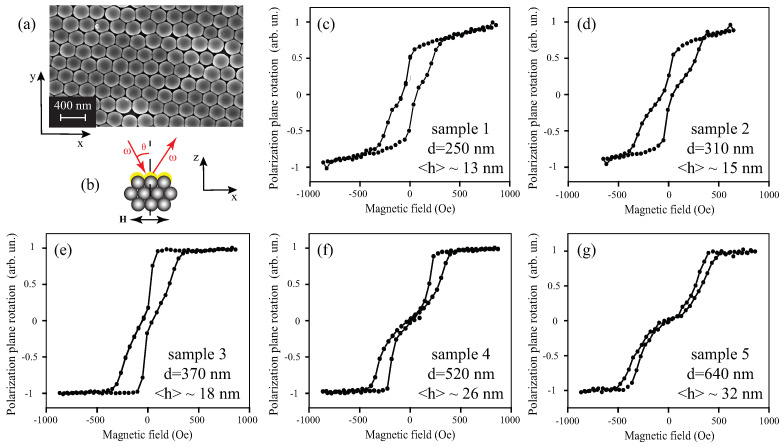
(**a**) SEM image of the surface of the Co-opal sample “1” with the diameter of the spheres *d* = 250 nm; (**b**) scheme of the longitudinal MOKE measurements (yellow color is for Co coating); (**c**–**g**) MOKE hysteresis loops for the samples “1”–“5”, respectively, the diameter of the spheres *d*, and the average thickness of cobalt are indicated on the panels.

**Figure 2 materials-14-03481-f002:**
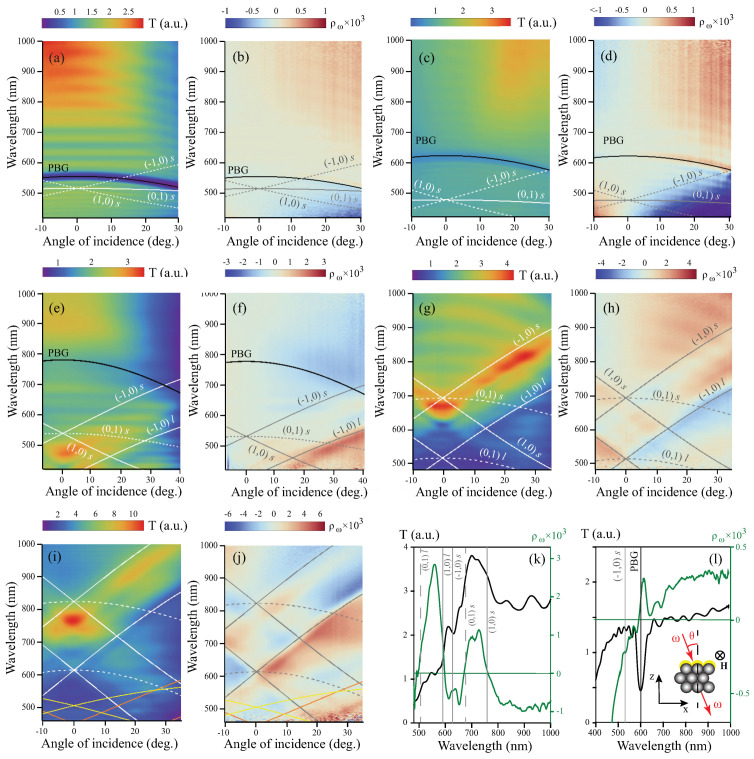
Angular-wavelength spectra of the transmission and magnetic contrast in the Voigt geometry for the samples “1” (**a**,**b**), “2” (**c**,**d**), “3” (**e**,**f**), “4” (**g**,**h**), and “5” (**i**,**j**); (**k**) T(λ) (black curve) and ρ(λ) (green curve) for the sample “4” at $\theta =-10^{\circ }$; (**l**) T(λ) (black curve) and ρ(λ) (green curve) for the sample “2” at $\theta =15^{\circ }$, inset: scheme of the experiment (yellow color is for Co coating).

## Data Availability

The data presented in this study are available on request from the corresponding author.

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
