# Peer review of "Size Effects in Optical and Magneto-Optical Response of Opal-Cobalt Heterostructures"

_materials, 2021, doi:10.3390/ma14133481_

Round 1

Reviewer 1 Report

The paper describes how to create composite structures that potentially could lead to efficient magnetoplasmonic structures by combining an opal film, that forms a photonic crystal, with perforated Co layers that support surface plasmon-polaritons. The periodically structured holes in Co are aligned with those in the opal structure, which leads to the well known anomalous transmission. In addition, the surface SPP excitation at the Co/opal interface is resonantly enhanced close to the photonic bandgap of the opal.

The paper gives a good overview of the state of the art: both magneto-plasmonic structures and magneto-photonic crystals have already been made and studied, but the combination is new and promises a combination of enhanced magneto-optical effects combined with anomalous transmission. However, unfortunately, the results demonstrate that the combined effects do not really deliver the desired strong response: an enhancement of 0.6% near the SPP excitation is found, while the combined effect of the PBG in the opal and the SPP in the Co even leads to a decrease in the MO response. By itself this is unfortunate, but not a reason not to publish this effort, as the idea looked good. However, before publication, the authors should address a number of issues to clarify the approach and obtained results.

  1. Hysteresis loops show a somewhat peculiar shape, with a "waist" near H=0, indicating a 2-step process, which is attributed to a vortex state magnetization. For the thinner two samples, the magnetization is not saturated above 500 Oe. The reference films are said to be uniformly magnetized in plane, but no loops are shown.
  2. The spectral dependence of both transmission and MO effects were measured with a magnetic field perpendicular to the structures (Voigt geometry). It is said that the applied field of 2kOe is enough to saturate the magnetization, referring to the MOKE results, but the latter are measured with H in plane. As Co shows in-plane magnetization, the perpendicular direction is the hard direction and it is not obvious that 2kOe is enough to saturate the magnetization in that direction. Why not showing MO results in that geometry?

    The spectral dependence of T is given in arbitrary units, which gives no indication wether these films really are having a stronger transmission. On the other hand, in the discussion later, actual numbers are mentioned that are consistent with those from the figures. So what is it: arbitrary or real numbers?

  3.  

    The measurements are compared with model calculations. It is said that the optical properties of the heterostructures can be analyzed via the different dependencies of the PBG and the positions of the SPPs, but this is not actually done in the paper.

  4.  

    Samples 1 and 2 are said to display only 1 spectral feature, and indeed show a clear minimum corresponding to the PBG. For the thicker samples, this is less pronounced.

    The V-like transmission maxima for structures 3-5 are associated with extraordinary transmission, due to the tunnelling of the light via the SPPs. These maxima are not observed for samples 1 and 2, which is said to be related to compensating effects due to the PBG. But why does this not result from the model calculations?

    In addition, for sample 1 there are also very distinct maxima and minima in the transmission as a function of wavelength above the PBG, in particular for negative angles of incidence. The maxima for sample 1 are as strong as for 3-5. Why is this not discussed? Also, why are these not symmetric around zero degree angle of incidence?

  5. The magnetic contrast in the used Voigt geometry is enhanced near the SPP. The maximum contrast is said to be 0.001, but it is unclear how much actual enhancement this is. In the abstract, an enhancement of up to 0.6% is mentioned, but I do not find this back in the paper. Either way, I do not think that a contrast of 0.001 can be very useful for applications, but maybe the authors can prove me wrong. The anomalous transmission is said to be up to 3.9 (p7), but the figure gives arbitrary units, so I cannot judge what this actually means. Also, how do these numbers compare with literature? 

    Why do the authors not study the Kerr effect, which is more straight forward for a metallic film? Is there no enhancement here? The Voigt effect would normally be zero and only gives an effect (linear in magnetisation) due to the breaking of symmetry.
  6. In summary: this is an interesting paper, demonstrating how to combine a photonic bandgap structure (opals) and surface plasmons (in metallic cobalt films), to enhance light-matter interaction and, in this case of a magnetic material, enhance magneto-optical effects. The idea is quite new, though the seperate effects of PBG and SPP on MO responses have been studied independently. It is not made very clear what the actual advantage is of the present structure, i.e. how it compares to the state-of-the-art. Also, the presentation of the results could be improved. Finally, as the authors also have done model calculations, they should at least come up with suggestions how these structures could be improved to get meaningful enhancements.           

  7. Some minor notes: The abstract starts with: Search for new types of efficient magnetoplasmonic structures that combine hightransparency with strong magnetooptical (MO) activity is very desired.

    Apart from the fact that this sentence is not grammatically completely correct, it is also not very clear to this referee why such a search would be very desirable.

    The paper could also profit from some critical proof reading.

Reviewer 2 Report

In the recent paper titled "Size Effects in Optical and Magneto-Optical Response of Opal-Cobalt Heterostructures" the authors present the size dependence of opal films with specific orientation (111) where a think cobalt layer was deposited.

While the combination of theoretical models with experimental data and the throrough analysis of their optical and magnetooptical properties is provided the authors have not provided characterization analysis for their synthesized heterostructures.

I believe that this manuscript is at the moment at an early stage. The authors need to provide convincing evidence of the purity of the synthesized material, the (111) orientation, more morphological characterization as well as elemental distribution and atomic percentage of the synthesized materials

Reviewer 3 Report

The manuscript entitled “Size Effects in Optical and Magneto-Optical Response of Opal-Cobalt Heterostructures” reports the use of photonic crystals based on silica opal crystal with Cobalt coating to study the magneto-optical properties in dependence of the silica nanospheres diameter. I recommend accepting this manuscript after a few minor revisions are implemented. The following points should be addressed.

Content editing:

  1. In line 1116-118, the authors mention that the morphology of the samples has been studied, however, a single image is reported (Fig1a) showing a single monolayer of SiO2 spheres. The image of the actual opal should be displayed.
  2. It would be good to spend a few words to delve deeper into the influence of the morphology of the opal crystals on the magnetization loops as well as the photonic band gap. In fact the self-assembly of nanospheres tend to produce grain boundaries. What is the effect of such grain boundaries on the MO properties? How did the authors account for the variability in the number of layers (in line 103 they state they have 3D ordered crystals with 10-15 stacked monolayers) ?
  3. What is the lateral resolution of the MOKE experiment with respect to the grain size?

Text editing:

  1. In the section Materials and methods, the authors refer to Figure 1 containing SEM image of the nanospheres as well as magnetization loops by calling it Figure 2, this should be corrected.
  2. In the caption of Figure 1, the authors write that the effective thickness of the cobalt layer is reported while in the panels they report the average thickness <h> which is the half of h_eff. This should be corrected as well.
  3. Figure 1 b would be more informative if tilted of 90° to the right, so that the z axis would be aligned vertically, while there’s no need to align the x axis as it is now between fig1a and fig1b because that axis is not actually aligned to the nanospheres across the substrate.

Reviewer 4 Report

This manuscript reported optical and magneto-optical properties of composite material comprising opal film and cobalt nanostructure perforated across it. It was reported that enhancement in both transmission of light and magneto-optical effects denoted by rho occurred and was accounted for by surface plasmon polariton excitation. Phenomenological description of the experimental results was given in a great detail, despite some typos such as those found in line 120. What follows is the list of issues that authors need to address or add to improve the manuscript.

  1. Authors need to provide more qualitative and insightful description of what is going on for SPP enhancement of magneto-optical anisotropy while allowing for high transmission of light.
  2. Additional schematic diagram for the experimental setup used had better be added with the direction of magnetic field applied.
  3. In addition to the rho parameters that were obtained by flipping the magnetic field polarity, authors had better address the magnetic hysteretic behavior of optical transmission under magnetic field loop for comparison with its hysteretic properties of MOKE presented in Figs.1(c)-(g).

The above mentioned issues need to be addressed in the revised manuscript before publication in the journal.       

Round 2

Reviewer 2 Report

Issues were addressed properly. The authors have provided sufficient characterization data